# Managing university e-learning environments and academic achievement in the United Arab Emirates: An instructor and student perspective

**Alberto Ibanez Fernandez**[1,2]* , **Ahmed Al Radaideh**[1], **Gyanendra Singh Sisodia**[3],
**Asok Mathew**[4], **Juan Antonio Jimber del Río**[5,6]

1 College of Business Administration, University of Science and Technology Fujairah, Fujairah, UAE,
2 International Researcher, Millersville University, Millersville, Pennsylvania, United States of America,
3 College of Business Administration, Ajman University, Ajman, UAE, 4 College of Dentistry, Ajman
University, Ajman, UAE, 5 Department of Social Sciences, University of Córdoba, Cordoba, Spain,
6 International Researcher, Universidad Ecotec, Guayaquil, Ecuador

☉ These authors contributed equally to this work.
* albertoif@yahoo.es

pone.0268338

CHINA

**Data Availability Statement:** All relevant data files
from this study are available from the Figshare

## Abstract

The present research evaluates how E-learning environment, E-learning adoption, Digital
readiness, and Students attitudes towards E-learning, affect Academic achievement. The
study focuses on a much-neglected cultural context, Gulf Cooperation Council countries
(GCC), since Student's readiness as well as institutions and professors' endowments
greatly varied within countries and among universities. The study further incorporates
Instructors attitudes and evaluates the mediation effect of Academic engagement on Aca-
demic achievement. The methodology relies on Partial Least Squares structural equation
modelling (PLS-SEM). The research findings emphasize the role of E-learning environment,
Digital readiness, Academic engagement, students as well as instructors E-learning attitude
as the decisive factors that determine students' Academic achievement. This implies that
institutions who adapt to a changing environment by aligning students and instructors' goals
to develop a positive and supportive E-learning environment, will foment Academic engage-
ment and promote students' Academic achievement.

## Introduction

The development of e-learning environments has been fostered by the nature of the interna-
tional pandemic context. Rather than focusing on the consequences of this international situa-
tion, the research analyses the potential variables that might foster academic achievement
within e-learning environments. Specifically, the study builds upon and goes beyond the exist-
ing literature, developing a model that takes into consideration six variables that include stu-
dents as well as instructors' perspective, along with the institutional endowments, and e-
learning environments. Based on the results, the study proposes policies and recommenda-
tions that might further enhance the efficiency and efficacy of e-learning environments.

database (DOI: 10.6084/m9.figshare.19403006.
v1).

**Funding:** The authors received no specific funding
for this work.

**Competing interests:** The authors have declared
that no competing interests exist.

The present research focuses on how to enhance students' e-learning experiences and academic achievement and evaluates the type of e-learning strategies and policies that will provide a more positive effect on students, within the present and post-pandemic context. Recent studies have explored the role of platforms such as Microsoft Teams, Zoom and Moodle on academic achievement during the Covid-19 pandemic [1]. Other recent studies address the potential relation between motivation and academic achievement in e-learning [2], e-learning readiness and academic achievement [3], and students' attitudes and academic achievement in e-learning [4]. One of the limitations of these studies that the present research attempts to address, resides in their individualized study of potential variables that affect academic achievement withing complex e-learning environments.

Building upon these and other results exposed along the literature review, the present research proposes a comprehensive model that explores the relevant individual variables tested along previous studies, its potential relations, as well as mediation effects, and includes students, as well as instructors' attitudes, to evaluate their effect on academic achievement, within e-learning environments.

The goal of this study consists in developing a comprehensive model to examine the potential relations between academic achievement and the variables under study, including students as well as instructor's perspectives, to develop e-learning environments that foster academic achievement.

## Literature review

Based on this goal, the present study examined prior findings, in order to develop the hypothesis and the proposed model. Previous studies have addressed students e-learning experiences and academic achievement [5–9]. E-Learning represents an opportunity to rethink the present educational framework, currently driven by market forces and educational organizational structures that must adapt to the present challenges and assure students' academic achievements.

One of the first measures implemented to cope with the global pandemic effects on education has been to transfer classes to an online environment. Nevertheless, we should understand the great differences between a well-planned online program and a quick fix to continue providing education. Institutions across the world are trying to draw conclusions about this online transfer process to better prepare for future educational frameworks. We could reach biased conclusions when analysing and comparing face-to-face learning with this emergency online experience. This could happen both directions. As an example, above average grades during this period might be presented as an endorsement of online education or as a lack of the proper supervision means during assessment periods. Instructors and students feedback related to their experience during the current times should provide the first component for further analysis and enhancement of online, as well as face-to-face teaching.

Pre-pandemic e-learning strategies in universities around the world have addressed and promoted the transition from an instructor focus approach to student´s centred experiences [10, 11], Student´s academic achievements have continued to increase during the past decade, based upon the enrichment of university e-learning environments [5, 6]. This might be due to a proper strategic planning process, which was not able to take place during the abrupt pandemic transition from in-person learning to e-learning. Study suggest that proper planning grants the opportunity to individualize materials and strategies [12]. This allows to increase student's engagement and digital readiness, providing a positive effect on student's academic achievements. Technological advancements in hardware, network capacity, software and audio and video communication applications and protocols foster e-learning experiences,

students' academic engagement and the implementation of new e-learning strategies [13]. Furthermore, the extended e-learning environment, such as e-learning platforms, online chats, virtual environments that simulate class scenarios, tools to share student screens and the recent capacities added to create private rooms within the e-learning classes, allow to implement in-person teaching strategies such as group cases, students presentations and a more immersive learning experience. All these newly available tools should enhance students' academic engagements and achievements although the literature review reflects contradictory results on the effect of e-learning on the students' engagement and academic achievement, students experience a higher level of satisfaction when engaged in e-learning [14]. A recent study reports a reduction in dropout rates and an increase in students grade point average (GDP) [15], while another study addressing similar phenomenon research results indicate an increase in critical-thinking skills for students involved in e-learning [16]. While these results are encouraging, a broader literature review also addresses the null or even negative relation between GPA and e-learning teaching [17], and the use of technology in the classroom and GPA [18].

Recent studies confirm the higher failure rates of e-learning students, despite being their first choice vs. traditional in-person teaching [19]. A cross sectional study during the Covid-19 pandemic reported a positive correlation between student´s technology addiction and their e-learning academic engagement [20]. Recent research explores students' acceptance and perceptions of e-learning during Covid-19 [21]. Their results indicate that students highly value the opportunity e-learning offers them to reconcile their personal and academic schedules and the time savings e-learning provides, mainly on logistics and commuting. These perceived advantages had a direct positive effect on student's academic engagement.

Despite the successful pre-pandemic e-learning experiences in different countries around the world, each country specific context should be taken into consideration prior to the establishment of policies to maximize their effectiveness [22]. The present research geographical focus is on the Gulf Cooperation Council country (GCC) of United Arab Emirates. There is a considerable lack of studies on e-learning for this area of the world, compared to Europe, North America, and other Asian countries such as China, India, and Japan. This research first contribution aims to help narrowing this gap. The present study second contribution aims at providing insight on the students e-learning experiences during this pandemic period. The research evaluates students' academic engagement, and digital background to implement policies and strategies that will preserve and foster students' academic achievements. The third contribution of the study consist in including within the model the perspective of the instructor, based on the results of a recent research adapted to the present model as Instructor´s Attitude [23]. The fourth and final contribution of the study is the inclusion of the latent variable E-learning environment [24–27].

The hypothesis for the following variables on the present study are based on the following studies. Studies reports a positive relation between Student's attitude towards e-learning, as well a student's digital readiness on academic achievement [28], while results of another study could not confirm these positive relations [29]. More recently, research results endorsed this positive relation [30, 31]. These mix results in the recent literature motivates the development of hypothesis three and nine for the present study.

Research reported mix results on the relation between academic engagement and academic performance [32], while another, concluded that the degree of academic engagement was a positive predictor of academic achievement [33], in a more recent study it was found that academic engagement, related to an increase of online assignments rushed to e-learning due to the pandemic circumstances, had a negative effect on students' academic achievement, especially for students who spent more time and effort on these assignments [34]. This might be due to the lack of meaningful content of certain activities, when transferred from an in person

to an online learning environment. The present research examines this relation and based on these mix results, explores its potential mediation role within the proposed context.

In terms of e-learning adoption and academic achievement, it was reported university students might or might not transfer their personal technological knowledge to the e-learning environment [35]. Recent findings indicate that there are cultural dimensions that affect the outcome of the relation. Due to these potential cultural effects [36], the present research includes its analysis for the geographical context into consideration.

The two proposed hypothesis for the potential mediation roles of digital readiness and academic engagement are derived from the future lines of research proposed in their study [8]. Finally, the present research includes within its hypothesis the potential relation between the instructor attitude and academic achievement within the e-learning context, based on the instructor characteristics proposed [37], to test the relation.

## Research model and hypotheses

The model is based on the literature review findings for the following factors: E-learning Attitude, Academic Engagement, E-learning Adoption, Digital Readiness, and Academic Achievement, and the proposed new factors within the model: Instructor Attitude and E-learning Environment, based on the literature findings.

The purpose of the present study is to investigate the effect of the aforementioned variables on Academic Achievement, with the inclusion of Instructor Attitude and E-learning contextual factors, to help develop e-learning environments that foster academic achievement. Furthermore, the study explores the potential indirect effects of the variables on Academic Achievement.

Fig 1 presents the research model and stablishes the respective hypothesis for each of the factors under consideration.

GPA represents a well stablished measure of students' Academic achievement [38]. All the students learning experience along four years of education is summarized in one number that determines student academic success. Due to this fact and based on previous research, GPA is stablished as a reasonable summary of students' Academic Achievement [39].

Researchers explored the effects of the pandemic on different country contexts [40]. It was reported differences in e-learning Adoption based on cultural factors [28, 41]. It was identified a positive relation between Academic Achievement and e-learning Adoption in South Korean higher education students [8]. Cultural contexts result vital when finding solutions to present and future educational challenges. Researchers developed a meta-analysis research to evaluate the relationship between student's engagement and academic achievement [31]. During the review, authors encounter conflicting results, although the conclusions established stronger evidence of positive relation between the two factors. Authors reported cultural values could further influence student's feedback.

Students e-learning Attitude represents a fundamental factor for their Academic success [42]. Researcher indicated that students who have a domain in web-based technology [43], experienced a positive e-learning Attitude during this pandemic period, while a Similar research study reported a 77% negative attitude towards e-learning among students from Liaquat College of Medicine in Pakistan [44]. This might be due to the different institutions and student´s technological endowments. factors such as accessibility, system quality, computer playfulness and computer self-efficacy were identified as critical for students to have a positive attitude towards e-learning [45].

Digital Readiness ".. can be one of the significant connections between the student's e-learning experience and Academic Achievement" [8] (p.6). Recent research identified the

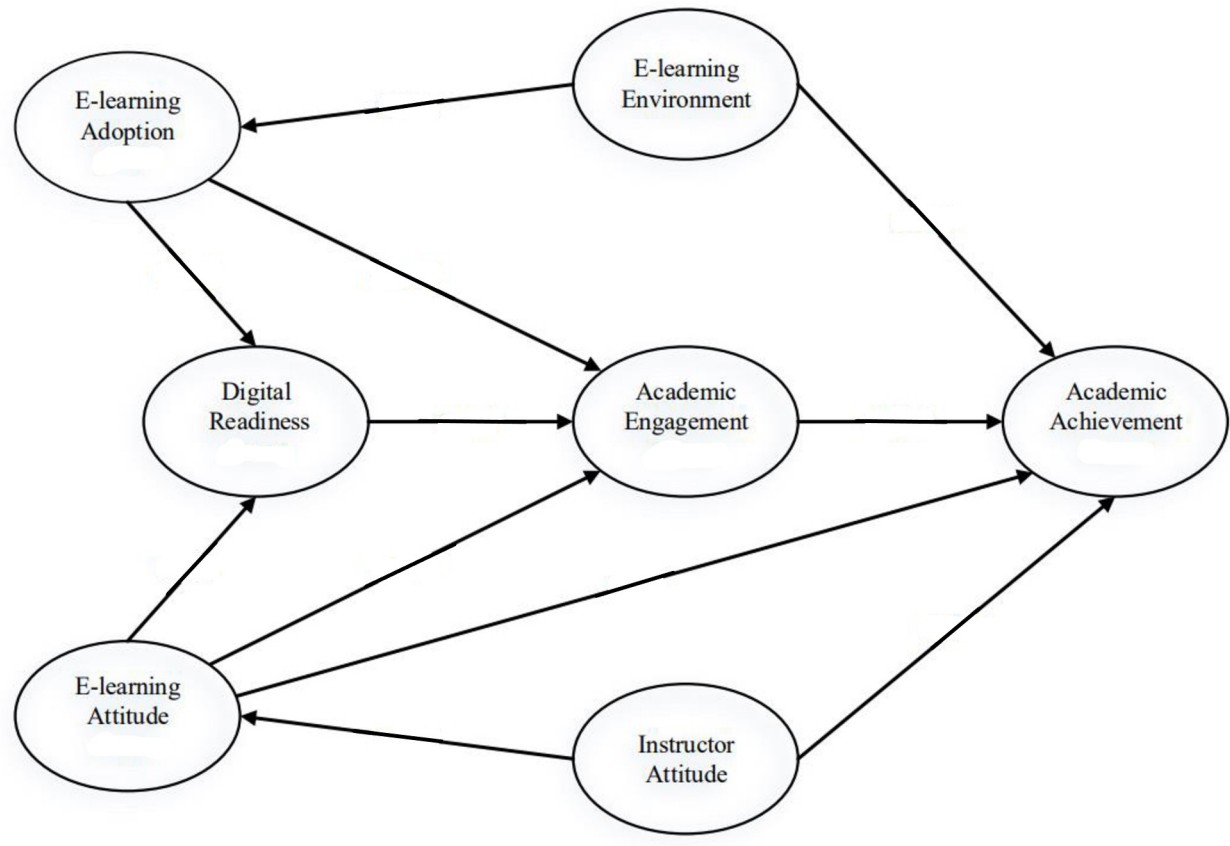

**Fig 1. Research model.**

availability of technology equipment for students, their previous exposures to blended or digital learning and individual technology skills, as key factors of students' Digital Readiness and potential precursors of academic Achievement [46].

The number of studies related to Instructors Attitude towards e-learning is still relatively small compared to the ones developed from the student´s perspective. To evaluate this effect, the present research included this factor within the model., Instructors Attitude towards e-learning represent a critical factor to guaranty student's satisfaction [47]. research has also identified the relation between Instructor´s Attitude and students' satisfaction within e-learning environments [13].

Based on the literature review and research goals, the present study tests the following hypothesis:

H1: Academic achievement is positively related to student´s e-learning adoption.

H2: Academic achievement is positively related to students' academic engagement.

H3: Academic achievement is positively related to students' e-learning attitude.

H4: Academic achievement is positively related to students' e-learning environment.

H5: Academic achievement is positively related to Instructor's attitude.

H6: E-learning environment is positively related with e-learning adoption.

H7: Academic engagement is positively related with e-learning adoption.

H8: Digital readiness is positively related with e-learning adoption.

H9: Academic engagement is positively related with digital readiness.

H10: Digital readiness is positively related with e-learning attitude.

H11: Academic engagement is positively related with e-learning attitude.

H12: Students e-learning attitude is positively related with instructor's attitude.

The present research further presents the potential indirect effects that might arise from the model under study.

## Methodology and data analysis

### Data collection

Data was collected from two different studies conducted for the undergraduate student E-learning fall semester 2020 experience at the University of Science and Technology Fujairah (USTF) in the United Arab Emirates. The two studies were conducted using an online questionnaire via Google Forms. Students were invited to participate voluntarily by email. Consent was informed, and obtained from the students within the survey, as well as verbal consent was granted by the university Ethics Committee and was witnessed by the Chair of the committee and two of the authors of the study during a regular Ethics Committee meeting. Students were asked to answer questions regarding their distance learning experience during their 2020 academic year.

### Data analysis and results

Data analysis was carried out using IBM SPSS 26 software alongside the Partial Least Squares structural equation modelling (PLS-SEM). The descriptive statistics for the collected demographic was done by using SPSS, while the model evaluation and path estimation performed by the PLS-SEM 3.0 software [48].

The regression-based method was used to explore the research model hypotheses using the latent variables with observed variables. The two-step approach of the first evaluation was used for the measurement model and the structural model [49–51]. Researchers used the PLS-SEM to understand the path coefficients and variance of the dependent variables which are explained by the independent variable in their suggested models [52, 53].

Preliminary analysis of the study model was performed on a randomized controlled trial using a fixed root means square remainder (SRMR). The model showed a value of 0.07, which was less than the recommended value (0.08), [54] This suggests that the research model fits well with the data. In addition, the appropriate full model indicator produces a value of 0.07 and confirms the positive value of the model equity [51].

Data was collected by using a sample size of 200 undergraduate students from USTF. The participant students were 55% Female and 45% Male, 54% of the participants' age were from category (17–22) and the overall average Age was (Mean = 25.02 and SD = 7.07). Table 1 shows the demographic characteristics of the sample.

Thirty one percent of them were from the college of humanities & science, and 21% were from the Dentistry department. The remaining 48% were from the rest of the university colleges. Thirty-six percent of participants were First-year students, and 11% were Fourth-Year students. A Likert scale was used to investigate and explore the off-campus E-learning environment. Eighty-five percent of respondents indicated they felt comfortable in the adaption of the

**Table 1. Demographic characteristics.**

| Variable | Category | Frequency | Percent |
|---|---|---|---|
| **Gender** | Female | 109 | 55 |
| | Male | 91 | 46 |
| **College** | Business Administration | 40 | 20 |
| | Dentistry | 41 | 21 |
| | Engineering and I T | 8 | 4 |
| | humanities and science | 62 | 31 |
| | Law | 22 | 11 |
| | pharmacy | 27 | 14 |
| **Academic year** | First year | 72 | 36 |
| | Second year | 57 | 29 |
| | Third year | 49 | 25 |
| | Fourth year | 22 | 11 |

E-learning. Eighty-two percent of students were using the internet to communicate with their classmates, and 84% of them had a good experience in using the digital learning resources. Finally, 76% considered digital learning as a rational initiative within the current pandemic context.

Researchers described the items that can be used to measure the student's perception regarding the E-learning Attitude and academic engagement, such as "Studying with E-learning is a good idea" and "All things considered, using the e-learning system is beneficial to me" [55]. Those items were considered in this study to measure the students' E-learning attitude. The collected data shows a high reliability as stated in Table 2. Cronbach's alpha was equal to 0.947.

Furthermore, students assessed their competencies when involved in E-learning based on [55] research. The item used to measure this latent variable was "I have the necessary knowledge for using the university e-learning system," "Using the university e-learning system is entirely within my control," and "I have the necessary resources for using the university e-learning system." The collected data indicated high reliability with Cronbach's alpha was equal to 0.868.

Researchers used Digital readiness to measure the student's digital competencies for Academic engagement [56]. Items used to measure Digital readiness in this study were highly reliable with Cronbach's alpha equal to 0.908. The scale was used to measure Academic Engagement which measures the behavioural efforts of the students [57]. Cronbach's alpha for this latent variable is 0.749, which provides a good reliability indicator.

E-learning Environment was measured by the efficiency of the technology used to allow students to communicate remotely, as well as the student's knowledge and ability to use of e-

**Table 2. Cronbach's alpha, composite reliability, and average variance extracted.**

| Constructs | Cronbach's Alpha | Composite Reliability | (AVE) |
|---|---|---|---|
| Academic Engagement | 0.749 | 0.836 | 0.562 |
| Digital Readiness | 0.866 | 0.908 | 0.711 |
| E-learning Adoption | 0.868 | 0.919 | 0.791 |
| E-learning Attitude | 0.947 | 0.966 | 0.905 |
| E-learning Environment | 0.912 | 0.956 | 0.916 |
| Instructor Attitude | 0.915 | 0.944 | 0.848 |
| Academic Achievement | 1.000 | 1.000 | 1.000 |

**Table 3. Discriminant validity analysis.**

| | (AVE) | Fornell Larcker | | | | | | | Heterotrait Monotrait Ratio (HTMT) | | | | | |
|---|---|---|---|---|---|---|---|---|---|---|---|---|---|---|
| LATENT VARIABLE | | AA | AE | DR | EA | ET | EE | IT | AA | AE | DR | EA | ET | EE |
| Academic Achievement (AA) | 1.000 | 1.000 | | | | | | | | | | | | |
| Academic Engagement (AE) | 0.562 | 0.135 | **0.750** | | | | | | 0.149 | | | | | |
| Digital Readiness (DR) | 0.711 | 0.046 | 0.629 | **0.843** | | | | | 0.056 | 0.719 | | | | |
| E-learning Adoption (EA) | 0.791 | 0.139 | 0.663 | 0.646 | **0.889** | | | | 0.150 | 0.776 | 0.720 | | | |
| E-learning Attitude (ET) | 0.905 | 0.105 | 0.640 | 0.617 | 0.776 | **0.951** | | | 0.108 | 0.686 | 0.655 | 0.830 | | |
| E-learning Environment (EE) | 0.916 | 0.224 | -0.089 | -0.054 | -0.102 | -0.003 | **0.957** | | 0.227 | 0.117 | 0.078 | 0.108 | 0.037 | |
| Instructor Attitude (IT) | 0.848 | 0.237 | -0.049 | -0.013 | -0.066 | 0.023 | 0.836 | **0.921** | 0.226 | 0.104 | 0.060 | 0.076 | 0.044 | 0.836 |

*The bold numbers in the diagonal row represent the square roots of the AVE.

learning technology. This underlying variable had a high level of reliability equal to 0.912. Instructor Attitude was measured using the following items: the instructor presented the material well and clearly, respect of the lecture time, and interaction between the instructor and students. Cronbach's alpha for this variable was equal to 0.915 indicating a high level of reliability.

## The measurement model

The reliability of the measurement model was evaluated using Cronbach's Alpha and the Composite reliability. Based on previous research, reliability values should be above 0.70. Cronbach's Alpha was between 0.749 and 0.947, suggesting strong evidence of reliability [58]. Researchers stated that the (AVE) values must be above 0.5 [59]. For the present model AVE values range was between 0.562 and 0.916

Tables 3 and 4 provide further proof for the discriminant validity. Loading for each item and its construct and cross loading on all the other constructs were evaluated. Each item has a

**Table 4. Matrix of loadings and cross-loadings of variables in the measurement model.**

| | | AE | DR | EA | EE | ET | IT |
|---|---|---|---|---|---|---|---|
| Academic Engagement | AE1 | **0.697** | 0.377 | 0.417 | 0.004 | 0.352 | 0.036 |
| | AE2 | **0.650** | 0.262 | 0.330 | -0.110 | 0.206 | -0.105 |
| | AE3 | **0.826** | 0.529 | 0.521 | -0.141 | 0.493 | -0.081 |
| | AE4 | **0.812** | 0.602 | 0.630 | -0.037 | 0.696 | -0.020 |
| Digital Readiness | DR1 | 0.530 | **0.816** | 0.644 | -0.106 | 0.597 | -0.082 |
| | DR2 | 0.587 | **0.864** | 0.606 | 0.011 | 0.658 | 0.002 |
| | DR3 | 0.456 | **0.858** | 0.423 | -0.010 | 0.340 | 0.052 |
| | DR4 | 0.521 | **0.834** | 0.450 | -0.075 | 0.410 | 0.004 |
| E-learning Adoption | EA1 | 0.518 | 0.506 | **0.881** | -0.069 | 0.623 | -0.044 |
| | EA2 | 0.605 | 0.618 | **0.930** | -0.088 | 0.761 | -0.043 |
| | EA3 | 0.634 | 0.588 | **0.855** | -0.111 | 0.674 | -0.088 |
| E-learning Environment_ | EE1 | -0.069 | -0.047 | -0.068 | **0.941** | 0.036 | 0.757 |
| | EE2 | -0.096 | -0.055 | -0.118 | **0.973** | -0.030 | 0.834 |
| E-learning Attitude | ET1 | 0.629 | 0.596 | 0.781 | -0.008 | **0.957** | 0.015 |
| | ET2 | 0.575 | 0.583 | 0.703 | 0.019 | **0.962** | 0.032 |
| | ET3 | 0.621 | 0.582 | 0.727 | -0.018 | **0.935** | 0.019 |
| Instructor Attitude | IT1 | 0.002 | 0.006 | -0.065 | 0.696 | 0.047 | **0.888** |
| | IT3 | -0.079 | -0.009 | -0.076 | 0.781 | -0.023 | **0.919** |
| | IT4 | -0.040 | -0.023 | -0.049 | 0.809 | 0.044 | **0.955** |

**Table 5. Hypotheses, path coefficients, and results.**

| | Path | Path Coefficients | T Statistics | P Values | Results |
|---|---|---|---|---|---|
| H1 | E-learning Adoption -> Academic Achievement | 0.165 | 2.327 | 0.032 | significant |
| H2 | Academic Engagement -> Academic Achievement | 0.102 | 4.990 | 0.000 | significant |
| H3 | E-learning Attitude -> Academic Achievement | 0.091 | 0.848 | 0.397 | Not significant |
| H4 | E-learning Environment -> Academic Achievement | 0.123 | 3.074 | 0.001 | significant |
| H5 | Instructor Attitude -> Academic Achievement | 0.152 | 2.363 | 0.032 | significant |
| H6 | E-learning Environment_ -> E-learning Adoption | 0.102 | 1.603 | 0.109 | Not significant |
| H7 | E-learning Adoption -> Academic Engagement | 0.292 | 2.524 | 0.032 | significant |
| H8 | E-learning Adoption -> Digital Readiness | 0.418 | 4.022 | 0.000 | significant |
| H9 | Digital Readiness -> Academic Engagement | 0.298 | 3.133 | 0.002 | significant |
| H10 | E-learning Attitude -> Digital Readiness | 0.293 | 3.371 | 0.001 | significant |
| H11 | E-learning Attitude -> Academic Engagement | 0.229 | 2.073 | 0.038 | significant |
| H12 | Instructor Attitude -> E-learning Attitude | 0.023 | 3.063 | 0.001 | significant |

higher loading within the construct than its cross-loadings. Table 3 shows two different criteria for evaluating the discriminant validity, the Fornell-Larcker criterion [59] and the Heterotrait-Monotrait ratio of correlations [51].

## Structural model

The path coefficient, coefficient of determination, and path significance were calculated using PLS-SEM. Table 5 shows the PLS-SEM results, which include the path coefficients estimates, T-values, and their P-values. 3000 resamples were used during the Bootstrapping to test the significance of the path coefficients.

T-values for each path were calculated and tested at the significance level of 0.05. Also, Fig 2 represents the model and the path coefficients. The results indicate that academic achievement is significantly positively related to student´s e-learning adoption with a P-value = 0.032 <0.05.

In addition, Academic achievement has a statistically positive relation to the student´s E-learning adoption since the P-value = 0.000, while the relationship between Academic achievement and the student's E-learning Attitude was not statistically significant (P-value = 0.397), which is greater than the 5% significant level.

Academic achievement was found to be significantly positively related to students' E-learning Environment (P-value = 0.001< 0.05). Similarly, Academic Achievement was significantly related to the instructor's attitude (P-value = 0.032< 0.05). E-learning Environment was not statistically significantly related to e-learning adoption (P-value = 0.109), whereas Academic Engagement was positively related to e-learning adoption with a significant P-value = 0.032. There was a significant positive relation between Digital readiness and E-learning adoption (P-value = 0.000< 0.05).

Digital readiness was significantly positively related to both Academic engagement and E-learning attitude. In addition, Academic engagement, and Academic achievement, as well as Academic engagement and student´s attitude are significantly positively related. Finally, student's e-learning attitude was significantly positively related to the instructor's attitude with a P-value = 0.001.

The PLS-SEM doesn't provide comprehensive indicators of the suitability of the models for examining the adequacy of the proposed models. Therefore, this structural model was assessed using structural path calculations and t-tests in addition to the explained variances $R^2$. As a standard basic testing for the structural model, it is proposed in the literature that the

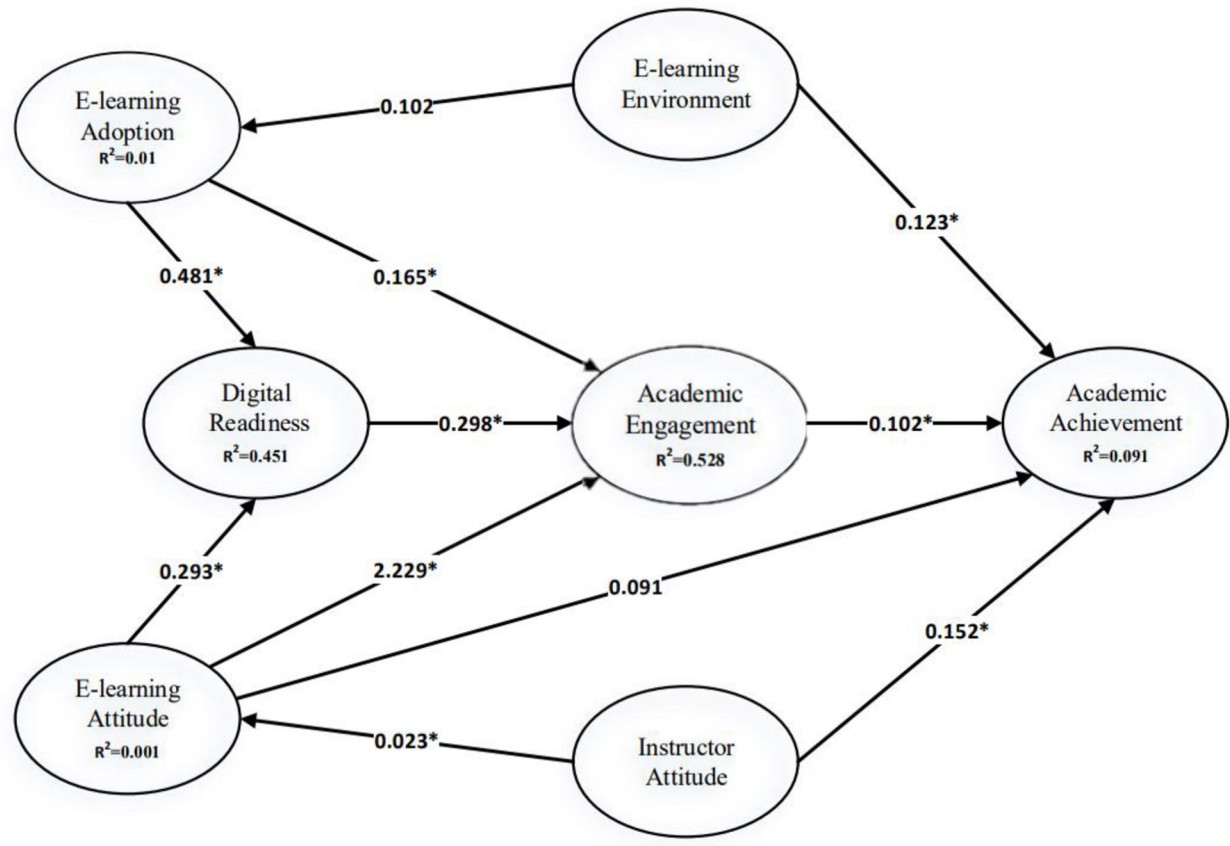

**Fig 2. Standardized path coefficients.**

coefficient of determination ($R^2$) can be used for the internal variables. The dependent construct, Academic achievement, had an $R^2 = 0.893$. Additionally, other constructs also had $R^2$ values for the following: Academic Engagement ($R^2 = 0.574$), Digital Readiness ($R^2 = 0.455$) and E-learning Attitude ($R^2 = 0.466$). Moreover, the Standardized Root Mean Square Residual (SRMR = 0.07) considered as a good model fit since its less than 0.10, [54].

## Discussion and implications

### Discussion

During the COVID-19 pandemic, all universities in the United Arab Emirates and most of the GCC countries, switched urgently from an in person direct teaching, to an online distance learning environment, due to the rapid spreading of the pandemic. Following Researchers proposed future lines of research, the present study evaluates the mediation role of Academic Engagement, expands previous models integrating variables related to instructors, applies the model to a specific cultural context to evaluate its ability for being generalized, and examines the results taking into consideration the present pandemic framework [8]. The discussion section contrasts previous findings with the present results and proposes policies to enrich e-learning environments.

A recent study examined students 'attitudes to assess their experience of the e-learning system during the first few weeks of the compulsory shift to online learning in the UAE during

the Covid-19 pandemic [60]. Their results indicated that the e-learning system is effective in terms of saving cost, time, and safety, during the period of the COVID-19 virus outbreak, while the negative results of the e-learning system included problems related to technology and insufficient technical support from teachers. These problems might have a direct effect on two variables under consideration in the present study, digital readiness, and instructor attitude. Another recent study investigated students' attitudes towards e-learning and virtual classes during the COVID-19 pandemic period [61]. The results established a statistically significant difference in students' attitudes towards the e-learning system and the level of students' satisfaction with the educational level of distance education [61]. As reflected in the literature review [13, 47], instructors' attitudes towards e-learning have a direct positive relation with student motivation. The present study new findings build upon these results by identifying the positive relation between instructors' attitudes and students e-learning attitudes, as well as on instructor attitudes and students' academic achievement. A recent study reinforces the relationship between attitude and engagement of students [8]. The following study [8] also proposed lines of research, the present study tested and observed the potential mediation effects of Academic engagement on Academic achievement. Academic engagement represents a mediator between student attitudes and academic achievement, as well as between student readiness and academic achievement. Academic engagement might play this role, if the transition of class activities to e-learning maintains a meaningful content and the activities are adapted to the new context [34]. The results on the positive relation between students' readiness and academic engagement, as well as engagement and achievement confirm the proposals of the literature review [12, 13]. The study further explores along the next lines, the present findings and relates them to previous studies within the literature review, as well as it proposes on the following section, actions that might enhance academic engagement to foster academic achievement.

Use of mobile technology enhances student's higher order thinking skills as well as it increases active engagement along the courses [9]. This might be due to the extreme familiarity of students with mobile technology. The results of the current study indicate that students must have confidence on their digital skills and commitment toward adopting E-learning by making the necessary efforts to learn and adapt to the new educational context. In that sense, self-learning represents a crucial element to attain Academic Achievement within the current scenario. Moreover, it might be recommended fostering students' participation in academic activities when designing the E-learning Environment, since the relationship between the E-learning system and students' Academic Achievement develops through academic participation. Consequently, it is relevant for the university to focus on supporting the E-learning process and community building to ensure that students have intensive training and experiences in using e-learning [62, 63].

The next relations to explore are related to Students e-learning adoption. The positive relation between Student's adoption and readiness, as well as adoption and engagement and adoption and achievement provide insight into the critical role Students adoption of e-learning play on their academic achievement. Achieving better academic results for students requires better Academic Engagement, and this will necessarily lead to a better e-learning environment at the university [64]. The process of making students more active through effective academic participation, leads to enhancing the cognitive and non-cognitive skills that students need to achieve academic success [16]. Previous studies indicated that e-learning attitude was considered more important than adopting e-learning. More recent studies support the findings of the present study on the critical role of Students adoption on academic achievement [8, 9].

The final variable under study is e-learning environment, which has a positive direct effect on academic achievement. As stated in the literature review, results are contradictory on the

relation of e-learning environments with academic achievement. This might be due to the definition of Academic environment adopted by each study, as it might focus from a classroom specific context to a broader individualized student context. The present research adopted the former perspective. The positive effect of current academic environments on academic achievements might be due to the availability of new tools such as e-learning platforms, virtual environments that replicate class scenarios, etc. . . which allow to transfer more effectively in-class activities to the new e-learning environments.

## Implications

Based on the current findings, the study presents university education leaders with strategies for an integrated approach to E-learning that provides practical implications. Universities might use teaching assistants as a university mentor to help students better use the electronic academic course portfolios. Furthermore, the figure of a teaching assistant allows the university to train students in different learning platforms and adjust their experiences within the E-learning environment, to improve performance and participation. The results also indicate that universities must provide the appropriate educational environment and infrastructure to enrich students' academic experiences. Fomenting students' interactions by providing individualizing guidance and support according to their educational file and personal context, will further enhance student's experiences by enriching educational engagement, which will lead to higher Academic achievement.

The recent reliance of universities on fully electronic education following the Covid-19 pandemic might make necessary for universities to implement seminars, workshops, and training sessions to enrich the learning environment, increase student's e-learning readiness and foster their academic engagement. These on-going training might also help foster students' positive attitude towards e-learning and further enhance their academic achievements by reducing their fear and anxiety promoted by the new learning environment. Conducting trial online exams will reduce student's apprehension and promote their confidence on themselves and on the E-learning environment. Training will further allow to foster the e-learning experiences of instructors and students, via their attitudes, having a direct positive effect on Academic Achievement. Parallel to these proposals, instructors might also be guided towards the urgent need to integrate technology into their courses and make the necessary arrangements to integrate the curricula into blended E-learning Environments. Furthermore, to foment positive instructors' attitudes towards e-learning, training sessions and continuous support must be ensured for faculty members.

As for the students, although younger generations are knowledgeable and to a certain extent dependent on technology, they need preparation and assistance to integrate academic work with these new technological skills [56]. This suggests that the implementation of intensive and effective educational activities to use the E-learning system helps to raise the students and faculty e-learning awareness levels. Moreover, faculty members must be aware of the E-learning systems, receive the appropriate training, and follow up on academic participation to raise students 'academic achievement [14].

As a final implication of the present study the results of this unplanned and unprecedented move towards e-learning due to the pandemic, should not overlook the potential benefits for students, faculty, and institutions. A well-planned transition from in person to e-learning can increase up to 60% students' academic material retention rates [65]. As a final consideration, the development and implementation of country general policies for e-learning will further provide a more certain and homogeneous academic context for students, instructors, and institutions. In that direction, the Ministry of Education (MoE) of the UAE developed

guidelines for all institutions regarding students and instructors code of behaviours for e-learning environments, in order eradicate cyberbullying and foment class interaction. Furthermore, the UAE MoE provides network access as well as computer equipment to all scholars. These policies ensure students accessibility to e-learning and provide guidance to students, instructors, and institutions and may be tailored and applied across countries and cultural settings.

## Conclusion, limitations, and future lines of research

The social, economic, and educational challenge that COVID-19 poses represent an opportunity to enhance current educational systems. The present study analyses fundamental factors to provide guidelines to improve students, instructors, and institutions e-learning experiences. The research findings emphasize the role of E-learning Environment, Digital Readiness, Academic Engagement, students as well as instructors E-learning Attitude and training, as the crucial factors that determine students' Academic Achievement. This implies that institutions who align students and instructors' goals to develop a positive E-learning environment, will foment Academic Engagement and promote students' Academic Achievement. One of the main limitations of the present research resides in the lack of control over students' backgrounds, personality traits, socio-economic environments, and educational backgrounds. This could represent an opportunity for future lines of research, as well as to extend the model to improve the findings on Instructors attitudes and apply the model in different cultural contexts. Further studies might benchmark students' results obtained prior to the pandemic context, with pandemic and post-pandemic e-learning environments. The results might allow to assess best practices from each learning environment, and develop educational models that align students, instructors, and institutions goals.

## Supporting information

**S1 Appendix.**
(DOCX)

## Author Contributions

**Conceptualization:** Alberto Ibanez Fernandez, Ahmed Al Radaideh.

**Formal analysis:** Ahmed Al Radaideh.

**Investigation:** Gyanendra Singh Sisodia.

**Validation:** Juan Antonio Jimber del Río.

**Writing – original draft:** Alberto Ibanez Fernandez.

**Writing – review & editing:** Asok Mathew.

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
