## [Decision Letter · Decision Letter 0]

21 Feb 2022

PONE-D-21-40699Managing University e-Learning Environments and Academic Achievement An Instructor and Student perspective.PLOS ONE

Dear Dr. Ibanez,

Thank you for submitting your manuscript to PLOS ONE. After careful consideration, we feel that it has merit but does not fully meet PLOS ONE’s publication criteria as it currently stands. Therefore, we invite you to submit a revised version of the manuscript that addresses the points raised during the review process.

Specifically, please consider the following during your revision:1) Please improve the introduction section with a proper description of the research background that emphasizes the importance of your research topic. Please justify the significance of your study with a brief summary of previous literature and their limitations, as well as the unique contribution of your study. (There are many research studies on online learning during Covid-19 pandemic, what makes your study necessary and unique?) 2) Consider revising your discussion section. First, please ensure that your discussion are based on the research findings. You can use sub-headings in discussion to make your argument more structured. Second, when citing other research studies in discussion, we try to link our results to previous findings (either in support or contradiction) or use previous literature to explain your own findings... I feel sometimes your reference lacks connection to your own results and sounds like part of general literature review (e.g., the second paragraph of discussion) Third, please consider using the first sentence of each paragraph as the summary of the paragraph content or topic of argument. The remaining sentences in the paragraph simply build upon the first sentence, providing elaboration, literature support or empirical evidence. I found in many occasions that the topic sentence is hided in the middle of paragraph or placed at the end of it.  (3) There are many related literature regarding online learning  during Covid-19, I believe the results can be used to compare and contrast your own findings, or provide insights for literature review. I list a few for your consideration.  reference 1: https://link.springer.com/article/10.1007/s10639-021-10791-xreference 2: https://link.springer.com/article/10.1007/s11516-021-0001-8

We look forward to receiving your revised manuscript.

Kind regards,

Heng Luo, Ph.D.

Academic Editor

PLOS ONE

Journal Requirements:

Reviewers' comments:

Reviewer's Responses to Questions

**Comments to the Author**

1. Is the manuscript technically sound, and do the data support the conclusions?

Reviewer #1: Yes

Reviewer #2: Yes

2. Has the statistical analysis been performed appropriately and rigorously? 

Reviewer #1: Yes

Reviewer #2: Yes

3. Have the authors made all data underlying the findings in their manuscript fully available?

Reviewer #1: No

Reviewer #2: Yes

4. Is the manuscript presented in an intelligible fashion and written in standard English?

Reviewer #1: Yes

Reviewer #2: Yes

5. Review Comments to the Author

Reviewer #1: Q1: The present research used Partial Least Squares structural equation modelling (PLS-SEM) to evaluate how E-learning environment, E-learning adoption, Digital readiness, and Students attitudes towards E-learning, affect Academic achievement. Based on the analysis results, the research emphasized the role of E-learning environment, Digital readiness, Academic engagement, students as well as instructors E-learning attitude as the decisive factors that determine students’ Academic achievement.

Q2: A total of 201 undergraduate students from USTF participated in the survey. The methodology relies on Partial Least Squares structural equation modelling (PLS-SEM) and the use of methodology is appropriate. The research presented Cronbach’s alpha, composite reliability, and average variance extracted to explain reliability and validity.

Q3: According to the author, the data was submitted to a public database, but he did not provide the URL of the database. In the section Data analysis and results, the study provides a descriptive analysis (including means, medians).

Q4: The manuscript presented in an intelligible fashion and written in standard English.

Reviewer #2: In the context of COVID-19, this study uses modeling to integrate teacher attitude variables and evaluate the mediating effect of academic engagement on academic achievement. The present research question focuses on how to preserve students’ traditional in-person academic achievements and educational experiences and evaluate the type of e-learning strategies and policies that will provide a more positive effect on students, within the present and post-pandemic context. Overall, the paper is well written and the data are analyzed comprehensively. However, the introduction seems to be a little simple and does not specify what the specific research question is. It would be better if the specific research question could be explained.

6. PLOS authors have the option to publish the peer review history of their article (what does this mean?). If published, this will include your full peer review and any attached files.

Reviewer #1: No

Reviewer #2: No

---

## [Author Response · Author response to Decision Letter 0]

27 Mar 2022

Answers to reviewer comments and improvements.

Thank you very much for all the suggestions and recommendations to enhance the present research. All of them will clearly enhance the quality of the present research.

**Introduction**

*The research background needed to be explained more clearly. What practical problem was the research aimed at solving? (Page 8, Lines 50 - 52).

Thank you very much for this very relevant point that helps clarify the nature and purpose of the study. The Introduction section has been intensively modified to clarify the aim of the study, as well as introducing the background needed.

* In this study, how is E-learning related to traditional learning? This sentence “The present research question focuses on how to preserve students’ traditional in-person academic achievements and educational experiences……” is too abrupt. (Page 8, Lines 59 - 61).

Thank you very much, the sentence has been eliminated, as it does not provide value, and replaced with a more profound explanation on the purpose of the present study. 

** Literature review**

The literature review summarized research on online learning or E-learning and elaborated the contribution of this study. However, the influencing factors mentioned in the study were not explained clearly. Specifically, despite stating the influencing factors, the study did not explain the relationship between factors and academic achievement. In the Research model and hypotheses section, the study mentioned “The model is based on the literature review findings for the following factors……”, but the arguments were not justified enough in the part of Literature review. 

Thank you very much, this is a critical element that has been addressed with the inclusion of new literature review that relates specifically to support the development of the proposed hypothesis. 

** Research model and hypotheses**

*Analyzing previous literature and making research hypotheses will help to clarify the relationships between variables.

Thank you very much, the introduction now includes background on the subject that will provide a better introduction to the literature review that supports the research hypotheses. Furthermore, the literature review has been expanded to include the studies that support the development of the proposed hypothesis. 

* Research hypotheses should be developed first, then a model should be developed to explain the relationship between the variables analyzed. The hypotheses should also be labeled on Figure 1.

Thank you very much, the development of the hypotheses, based on the literature review, to finally generate a model, has been described in detail. The hypotheses also have been labeled on Figure 1.

**Methodology and Data Analysis**

*A descriptive statistical analysis is best presented as a table.

* An analysis of model fit is missing from this study.

Thank you very much. The descriptive statistical analysis has been presented as a table and the analysis of model fit is included.

** Discussion and implications**

The discussion section is too weak. The arguments needed to be focused on results and elaborated with more studies/research. 

Thank you very much for these very relevant aspects to improve the discussion section. 

The results have been linked to the literature review, which also has been extended.

The discussion has been organized in two different sections to improve the structure.

Each paragraph adds a connection to the next to build upon the first sentence. 

Related literature regarding online learning during Covid-19 has been extended in order to compare and contrast the research findings. 

Specifically, please consider the following during your revision:

1) Please improve the introduction section with a proper description of the research background that emphasizes the importance of your research topic. Please justify the significance of your study with a brief summary of previous literature and their limitations, as well as the unique contribution of your study. (There are many research studies on online learning during Covid-19 pandemic, what makes your study necessary and unique?)

Thank you very much for this very relevant point. The Introduction section has been intensively modified to clarify the aim of the study, as well as introducing the background needed with a summary of previous literature and their limitations. Contributions are made more explicit also before the hypothesis are stated.

2) Consider revising your discussion section.

First, please ensure that your discussion are based on the research findings. You can use sub-headings in discussion to make your argument more structured.

Second, when citing other research studies in discussion, we try to link our results to previous findings (either in support or contradiction) or use previous literature to explain your own findings... I feel sometimes your reference lacks connection to your own results and sounds like part of general literature review (e.g., the second paragraph of discussion)

Third, please consider using the first sentence of each paragraph as the summary of the paragraph content or topic of argument. The remaining sentences in the paragraph simply build upon the first sentence, providing elaboration, literature support or empirical evidence. I found in many occasions that the topic sentence is hided in the middle of paragraph or placed at the end of it. 

(3) There are many related literature regarding online learning during Covid-19, I believe the results can be used to compare and contrast your own findings, or provide insights for literature review. I list a few for your consideration. 

reference 1: https://link.springer.com/article/10.1007/s10639-021-10791-x

reference 2: https://link.springer.com/article/10.1007/s11516-021-0001-8

Thank you very much for these very relevant aspects to improve the discussion section. 

The results have been linked to the literature review, which also has been extended.

The discussion has been organized in two different sections to improve the structure.

Each paragraph adds a connection to the next to build upon the first sentence. 

Related literature regarding online learning during Covid-19 has been extended in order to compare and contrast the research findings.

---

## [Editor Report · Decision Letter 1]

18 Apr 2022

PONE-D-21-40699R1Managing university e-learning environments and academic achievement in the United Arab Emirates: An instructor and student perspective.PLOS ONE

Dear Dr. Ibanez,

Thank you for submitting your manuscript to PLOS ONE. After careful consideration, we feel that it has merit but does not fully meet PLOS ONE’s publication criteria as it currently stands. Therefore, we invite you to submit a revised version of the manuscript that addresses the points raised during the review process.

I appreciate the extensive revision efforts taken by the authors. As a result, the quality of the paper has been significantly increased, and I don't see any major issues regarding the paper content.  However, before I can accept this paper, the authors need to further refine the format of the manuscript to meet Plos One requirements. The guidelines and sample paper of Plos One are attached in this e-mail for your reference.  Also, The color and style of proposed model in Figure 1 and validated model in Figure 2 are drastically different. I would suggest make them consistent in design and style. 

We look forward to receiving your revised manuscript.

Kind regards,

Heng Luo, Ph.D.

Academic Editor

PLOS ONE 
---

## [Author Response · Author response to Decision Letter 1]

26 Apr 2022

I appreciate the extensive revision efforts taken by the authors. As a result, the quality of the paper has been significantly increased, and I don't see any major issues regarding the paper content. 

Thank you very much for all the suggestions and recommendations to enhance the present research. All of them will clearly enhance the quality of the present research.

However, before I can accept this paper, the authors need to further refine the format of the manuscript to meet Plos One requirements. The guidelines and sample paper of Plos One are attached in this e-mail for your reference. 

The format of the manuscript has been adapted to meet Plos One requirements, based on the guidelines and sample paper.

Also, The color and style of proposed model in Figure 1 and validated model in Figure 2 are drastically different. I would suggest make them consistent in design and style. 

The color and style of proposed model in Figure 1 and validated model in Figure 2 have been redesigned to present a consistent style. 

Thank you again for all the support and guidance along the entire process. We look forward to provide value to the journal with the present research.

Very kind regards,

---

## [Editor Report · Decision Letter 2]

28 Apr 2022

Managing university e-learning environments and academic achievement in the United Arab Emirates: An instructor and student perspective.

PONE-D-21-40699R2

Dear Dr. Ibanez,

We’re pleased to inform you that your manuscript has been judged scientifically suitable for publication and will be formally accepted for publication once it meets all outstanding technical requirements.

Kind regards,

Heng Luo, Ph.D.

Academic Editor

PLOS ONE
---

## [Editor Report · Acceptance letter]

4 May 2022

PONE-D-21-40699R2 

Managing university e-learning environments and academic achievement in the United Arab Emirates: An instructor and student perspective. 

Dear Dr. Fernandez:

I'm pleased to inform you that your manuscript has been deemed suitable for publication in PLOS ONE. Congratulations! Your manuscript is now with our production department. 

Kind regards, 

on behalf of

Dr. Heng Luo 

Academic Editor

PLOS ONE